# Compound Eye Structure and Phototactic Dimorphism in the Yunnan Pine Shoot Beetle, *Tomicus yunnanensis* (Coleoptera: Scolytinae)

**DOI:** 10.3390/biology14081032

**Published:** 2025-08-11

**Authors:** Hua Xie, Hui Yuan, Yuyun Wang, Xinyu Tang, Meiru Yang, Li Zheng, Zongbo Li

**Affiliations:** 1Key Laboratory of Forest Disaster Warning and Control in Yunnan Province, Southwest Forestry University, Kunming 650224, China; hxie21527@outlook.com (H.X.); yuanhui314@outlook.com (H.Y.); insectyuyunw@outlook.com (Y.W.); txy8824@outlook.com (X.T.); meiruyang@outlook.com (M.Y.); 2College of Ecology and Environment, Southwest Forestry University, Kunming 650224, China

**Keywords:** bark beetle, *Tomicus yunnanensis*, apposition eye, open rhabdoms, vision, UV

## Abstract

The bark beetle *Tomicus yunnanensis*, a major forest pest in southwest China, primarily uses smells, but also sight (like seeing tree color changes), to find host trees. This study investigated its eyes and vision. We found the beetle has compound eyes made of 224–266 ommatidia, with the left and right eyes slightly different. The top third of the eye has square-shaped facets, while the rest are hexagonal. Each ommatidium has a large lens, a cone below it, and retinular cells forming a central core (R7/R8) and surrounding ring (R1–R6). Pigment cells surround these units. The eyes adapt to light and dark by changing the cone and rhabdom (without pigment moving), controlling how much light enters. Beetles fly only during the day, most actively between 7 and 11 AM. Light attraction tests showed females were strongly drawn to ultraviolet (360 nm and 380 nm) and red (700 nm) light. Males are highly sensitive to ultraviolet (360 nm) and violet (400 nm) light. This research helps us understand how these beetles see and how they combine vision with smell to find hosts and avoid other trees.

## 1. Introduction

The Yunnan pine shoot beetle, *Tomicus yunnanensis* Kirkendall & Faccoli (Coleoptera: Curculionidae, Scolytinae), was initially misidentified as *T. piniperda* following its first damaging outbreaks on *Pinus yunnanensis* in southwest China during the 1980s [1,2]. Over the past three decades, it has become the region’s most notorious forest pest, responsible for destroying 1.5 million hectares of Yunnan pine forest in Yunnan province alone [3,4]. Like congeneric species, *T. yunnanensis* exhibits a distinct annual life cycle with two phases: shoot-feeding and trunk-breeding [2,5]. New adults emerge in April–May, influenced by local climate and altitude. They migrate to healthy pine crowns and bore into the season’s new shoots for maturation feeding. This shoot-feeding phase lasts 1.5–3 months (ca. 3–5 shoots per beetle), causing significant shoot pruning, which weakens trees. Mass assemblies of beetles on shoots during this phase collectively predispose trees to subsequent attack [3,6]. Matured adults then disperse to seek suitable breeding materials, initiating the trunk-breeding phase by boring through the bark into the phloem, primarily in the stem’s middle or upper sections. After mating, females construct longitudinal egg galleries beneath the bark; hatching larvae feed on phloem and cambium. Larvae develop through three instars before pupating in chambers at tunnel ends. New adults emerging from pupae feed briefly on nearby phloem before boring exit holes. These adults typically return to living pine crowns for shoot feeding, initiating a new life cycle. A critical factor in *T. yunnanensis* impact is its relatively low attack density threshold for killing *P. yunnanensis*, estimated at around 80 attacks/m^2^ [3]. This threshold is notably lower than those reported for related species: *T. piniperda* on *P. sylvestris* (~300 attacks/m^2^) [7], *T. destruens* on *P. halepensis* (~178 attacks/m^2^) [5], *Ips typographus* on healthy *Picea abies* (200–400 attacks/m^2^) [8], and *Dendroctonus ponderosae* on *P. contorta* (50–120 attacks/m^2^) [9]. This comparatively low threshold required to overcome tree defenses, combined with its predisposition strategy during shoot-feeding, explains the frequent and severe damage caused by *T. yunnanensis* in southwest China.

*Tomicus yunnanensis* adults primarily locate hosts using chemical cues: host terpenoids (e.g., α-pinene, 3-carene, and 2-thujene) and aggregation pheromones (e.g., (-)-trans-verbenol and verbenone) emitted from initial galleries [10,11]. However, the potential role of vision in host location remains largely unexplored. Crucially, fundamental data on the species’ visual anatomy (including compound eye structure) and light-guided behavioral responses are lacking, despite evidence supporting visual capabilities in other Scolytinae beetles [12,13,14,15,16,17,18]. These beetles exhibit visual orientation, being attracted to dark surfaces (e.g., black, brown, red) and averse to white/yellow (negative phototaxis), even with attractants present [14,19,20]. Species like *Dendroctonus pseudotsugae*, *D. frontalis*, *D. brevicomis*, *I. paraconfusus*, and *Trypodendron lineatum* possess photoreceptors maximally sensitive to blue (~450 nm) and green (~520 nm) light (likely including UV), explaining photopositive responses to these wavelengths [21,22,23,24,25]. Final host selection involves assessing site characteristics (e.g., form, color, and texture) within the pine crown, related to resource availability [26], where visual cues may aid spatial assessment and enhance colonization. Crucially, our observations confirm *T. yunnanensis* exhibits strictly diurnal flight activity, leaving natal trees and boring into young shoots exclusively during daylight hours.

Furthermore, our group’s preliminary results indicate that the limited effectiveness of chemical lures alone in field trapping suggests olfactory cues are insufficient. These findings strongly suggest that visual capabilities play an indispensable role in the final approach and landing phases [27,28].

To characterize the visual system of adult *T. yunnanensis*, we (1) analyzed compound eye structure using multiple microscopy techniques, (2) recorded daily activity rhythms through behavioral observation, and (3) assessed spectral sensitivity via phototaxis bioassays. These findings enabled characterization of their visual physiology and comparative analysis with related species. Our study demonstrates how integrated visual–olfactory cues guide host location, informing targeted pest control strategies [29].

## 2. Materials and Methods

### 2.1. Field Collection and Laboratory Sample Preparation

Adult specimens of *T. yunnanensis* were collected alive within pine stems or shoots during January–February from the JiuLong Shan Forestry Station (25°0′35″ N, 103°7′15″ E), located in central Yunnan Province, Southwest China. Trees in the early stages of infestation or predisposed to attack were selected based on shoots and trunks exhibiting abundant boreholes and resin exudation, crown dehydration, progressive wilting, and near-total shoot occupancy by *Tomicus* beetles (representative symptoms illustrated in Cui et al. [6]). After felling, infested stems and shoots containing *Tomicus* beetles were placed in nylon mesh bags (50 × 100 cm, 80-mesh) and transported to our laboratory in Kunming for morphological and behavioral observations. Following the method described by Lieutier et al. [5], the key diagnostic traits used to distinguish *T. yunnanensis* from the other two sympatric species, *T. minor* and *T. brevipilosus*, primarily concern the characteristics of the second interstria on the declivity, the elytral vestiture, and the presence of erect interstrial hairs. The sex of *T. yunnanensis* specimens was systematically distinguished based on sound production during stridulation: sound producers were male; non-producers, female. To guard against misclassification, sex was further validated by examining the shape of the last abdominal tergites (arch in females vs. rectangle in males) [30] using a Zeiss Discovery V20 stereomicroscope (20× magnification) (Zeiss, Jena, Germany), especially when beetles were silent.

To conduct photoadaptation experiments, beetles were exposed to either high-intensity LED light (1000 lux) for at least 2 h or kept in complete darkness for at least 24 h within a purpose-built soundproof laboratory chamber (2 m × 2 m × 3 m). Light intensity was measured using a UT383S illuminometer (UNI-T, Shenzhen, China). Light-adapted beetles were decapitated and fixed in daylight, while dark-adapted beetles were processed under dim red light.

### 2.2. Scanning Electron Microscopy (SEM)

For compound eye morphology examination, detached *T. yunnanensis* heads (10 males, 15 females) were fixed in 2.5% glutaraldehyde-paraformaldehyde in 0.1 M phosphate-buffered saline (PBS, pH 7.4) at 4 °C for 12 h. After ultrasonic cleaning using standard parameters in a commercial-grade cleaner (UC-505, Dretec, Osaka, Japan), samples were dehydrated in a graded ethanol series (70%, 80%,90%, 95%, and 100% alcohol), 30 min per step, with three 100% ethanol changes. Following critical point drying, samples were mounted on stubs with conductive tape overnight. Gold coating was applied using a JFC-1600 sputter coater (JEOL, Tokyo, Japan) before observation/photography with a HITACHI S-4800 SEM (JEOL, Tokyo, Japan) at 15 kV. One specimen was sagittally sectioned along its head midline with a fine blade to map facet distribution and surface area.

### 2.3. Transmission Electron Microscopy (TEM)

For compound eye internal structure examination, heads were detached and immediately fixed in 2.5% glutaraldehyde and 2% paraformaldehyde in 0.1 M PBS. To enhance fixation and prevent degradation, degassing was performed: samples were placed in a syringe barrel, the needle hub sealed with parafilm, and vacuum maintained for 30 min after plunger retraction. Fixed samples were stored overnight at 4 °C. Subsequently, samples were washed thrice with ddH_2_O (7 min per wash), dehydrated through a graded ethanol series (as described in the SEM protocol), treated with acetone for 5 min, then embedded in Epon 812 resin and polymerized at 60 °C for 48 h. Using a Leica EM UC7 ultramicrotome (Leica, Wetzlar, Germany), uniform serial sections were prepared including semithin sections (800 nm thickness) and ultrathin sections (60 nm thickness). Ultrathin sections on copper grids were double-stained with 2% uranyl acetate and lead citrate. Samples were imaged using a JEM-1400Plus TEM at 80 kV (JEOL, Tokyo, Japan).

### 2.4. Light Microscopy (LM)

For the resin-embedded TEM samples, 200 nm-thick sections were cut using a Leica EM UC7 ultramicrotome. These sections were stained with 1% toluidine blue solution on a hot plate for 100 s. After cooling, sections were mounted with neutral balsam, labeled, and stored in slide boxes. Finally, samples were observed and photographed under a BX-51 fluorescent microscope (Olympus, Tokyo, Japan).

### 2.5. Observation of Daily Behavioral Rhythms

Daily activity of *T. yunnanensis* was observed in a laboratory under ambient room temperature and humidity conditions in Kunming during March, coinciding with their natural peak emergence period from pine stems. Observations spanned 24 h at 60 min intervals, with a defined photoperiod: light phase (07:30–19:30) and dark phase (19:30–07:30). Dark phase behavior was monitored under dim red light. We quantified insect activity by recording the number of *T. yunnanensis* adults emerging from stems at nylon bag entry points during each interval. Three replicate bags (each containing identical stem quantities) were monitored simultaneously. After each recording, all beetles were removed from the bags.

### 2.6. Testing Phototaxis with Monochromatic Light Across Multiple Wavelengths

The phototaxis behavior of *T. yunnanensis* was evaluated under discrete wavelengths using a three-chamber apparatus modified from Kim’s design [31]. Monochromatic light across the known visual sensitivity range of scolytid beetles (300–700 nm, 20 nm increments) [15,22,24,25] was generated by a CME-Mo151 monochromator (Microenerg, Beijing, China) with intensity fixed at 100%; irradiance at the stationary zone was calibrated to 300 lux through source-distance adjustments. Prior to testing, separating boards were inserted to delineate light, dark, and stationary zones. Test beetles were transferred to the stationary zone, covered for 10 min dark adaptation, after which the light source was activated while slowly withdrawing the boards. The apparatus was then covered with a black cloth to minimize interference. Following a 10 min exposure period, beetles in all zones were counted. Between trials, chamber walls were wiped with 95% ethanol, including optionally accelerated by hairdryer use, with subsequent trials initiated only after complete evaporation. For each wavelength, 30 newly emerged same-sex adults were randomly selected for one trial. Three replicates per trial were conducted across different sexes. Each beetle is used exclusively once for phototaxis evaluation. All experiments occurred in a self-built darkroom.

### 2.7. Morphometric Data Processing and Analysis

Body lengths of *T. yunnanensis* were measured using a Zeiss Stereo Discovery V20 (Zeiss, Jena, Germany). SEM micrographs quantified histologic parameters: dorsoventral height, anterior-posterior width, ommatidia count, facet diameters, and facet areas. Longitudinal LM sections measured ommatidial length, eye radius of curvature, and corneal facet dimensions, as well as interommatidial angles. The radius of curvature was measured at the broadest anterior-posterior point using a circular tool on the eye surface using image analysis software. Transverse LM sections measured rhabdom cross-section areas. TEM micrographs assessed the number of corneal laminations and retinular cells, diameters of pigment granules and mitochondria; and determined cone length, corneal thickness, and pigment granule morphology/position within retinula cells. Retinular cell numbering followed Wachmann [32], and rhabdom pattern classification used Schmitt et al.’s leaf beetle criteria [33]. Eyes from both sexes were examined via LM and TEM, with at least three replicates per sex/treatment group. For adult *T. yunnanensis*, six key optical parameters (focal length, image focus length, F-number, ommatidial acceptance angle, interommatidial angle, and eye parameter) were calculated from anatomical measurements [34,35].

All morphometric data were acquired using ImageJ software (Version 1.53, National Institutes of Health, Bethesda, MD, USA), measuring an identical number of objects per specimen to ensure consistent sampling and prevent statistical bias. Statistical differences between sexes or phototaxis groups were analyzed using unpaired *t*-tests and LSD post-hoc tests following ANOVA in R (RStudio). Figures were created using Adobe Photoshop CC 2019 (Adobe, San Jose, CA, USA).

## 3. Results

### 3.1. Gross External Morphology of the Eye

Adult *T. yunnanensis* specimens collected in the field ranged in body length from 5.7 mm to 8.4 mm (Table 1). The compound eyes are bilaterally symmetrical, dark in coloration, and fully oval-shaped, elongated along the dorsoventral axis (Figure 1A–C). Notably, they lack a small projection of the head capsule dividing the eye center, which differs from other scolytinae beetles. The eyes lie flush with the surrounding head structures. Their height is 576.3 ± 39.0 µm in males and 593.3 ± 27.6 µm in females, while their width is 219.3 ± 13.0 µm in males and 220.6 ± 14.4 µm in females (Table 1). No significant correlation was found between body size and eye size in either males (Pearson correlation, r = 0.356, *p* = 0.199) or females (r = 0.391, *p* = 0.167). Each eye comprises numerous compact ommatidia, featuring smooth, slightly convex surfaces and interommatidial setae (ommatrichia) (Figure 1C–E). Males possessed an average of 240.00 ± 3.51 facets per eye, while females had 247.38 ± 3.18 facets per eye. Facets within the dorsal third of the eye are predominantly quadrilateral in shape (Figure 1C,D). Facets elsewhere are typically regular hexagons, although some facets towards the peripheral region exhibit irregular shapes (Figure 1C,E). The surface area of quadrilateral facets (363.88 ± 3.19 µm^2^) was significantly smaller than that of hexagonal facets (406.67 ± 2.28 µm^2^) (*p* < 0.0001). While the external morphology of the left and right compound eyes was identical, the number of facets and their surface area distribution differed (Figure 2).

**Figure 1 biology-14-01032-f001:**
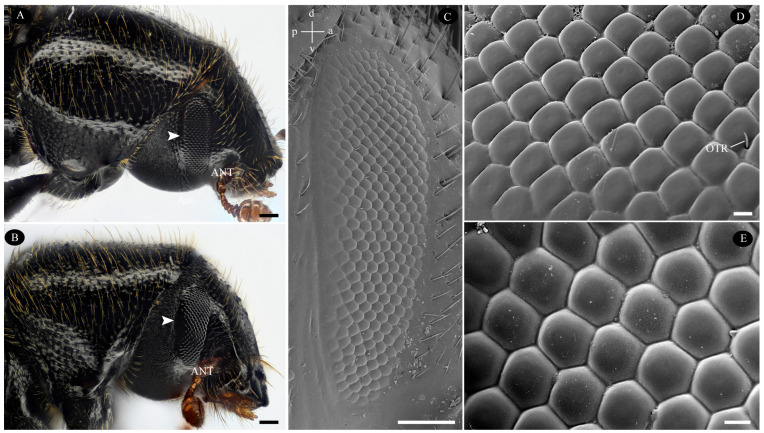
External morphology of the compound eyes in the *Tomicus yunnanensis* adult. (**A**) Position of compound eye in a female (arrowhead); (**B**) Position of compound eye in a male (arrowhead); (**C**) The eyes exhibit a fully oval-shaped morphology, with sparse ommatrichia primarily localized in the mid-inferior region (a—anterior; p—posterior; v—ventral; d—dorsal). (**D**) Quadrilateral ommatidia are predominantly found in dorsal third view. (**E**) Hexagonal ommatidia are visible in mid and ventral views. ANT—antennae; OTR—ommatrichia; Scale bar: (**A**–**C**) = 100 µm; (**D**,**E**) = 10 µm.

**Figure 2 biology-14-01032-f002:**
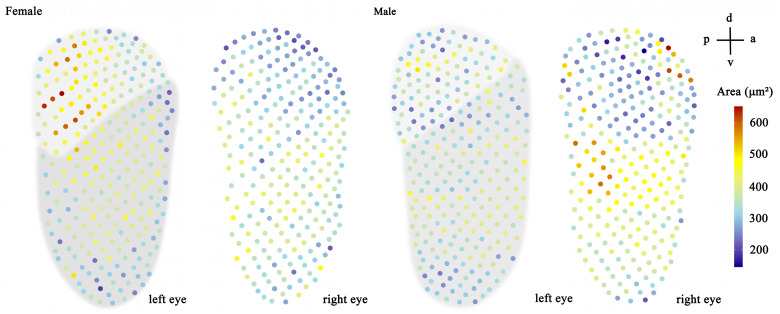
Facet size and distribution in female and male *T. yunnanensis*. The dorsal region (low-gray) of the left eye exhibits quadrilateral ommatidia, while the ventral region (high-gray) shows hexagonal ommatidia. Anatomical orientation is indicated top-right (a—anterior; p—posterior; v—ventral; d—dorsal).

**Table 1 biology-14-01032-t001:** Histological and optical parameters in the compound eyes of *T. yunnanensis*.

Structural Elements	Parameters	Unit	*n*	Average	Range (Min–Max)
Body size	Males	mm	30	7.4 ± 0.5	5.7–8.4
	females	mm	30	7.1 ± 0.5	6.5–8.1
Compound eyes	Eye height in males	µm	30	576.3 ± 39.0	518.2–641.3
	Eye height in females	µm	30	593.3 ± 27.6	528.0–644.9
	Eye width in males	µm	30	219.3 ± 13.0	198.5–241.6
	Eye width in females	µm	30	220.6 ± 14.4	201.2–260.2
	Facet number in males	-	14	240.0 ± 3.5	224–265
	Facet number in females	-	16	247.4 ± 3.2	224–266
	Eye radius in males	µm	12	481.5 ± 53.9	417.5–595.8
	Eye radius in females	µm	12	481.9 ± 45.6	412.1–571.3
ommatidia	Facet diameter	µm	130	21.0 ± 2.9	13.5–25.8
	Quadrilateral facet area	µm^2^	150	363.9 ± 3.2	278.7–481.9
	Hexagonal facet area	µm^2^	150	406.7 ± 2.3	358.4–501.7
Cornea	Maximum thickness	µm	10	20.7 ± 1.1	19.2–22.3
	Thickness of OLU	µm	30	16.0 ± 2.4	12.4–18.7
	Thickness of ILU	µm	30	2.4 ± 0.5	1.3–3.4
	Number of chitin layers	µm	5	69.6 ± 1.2	67–74
	Outer lens surface radius	µm	23	17.4 ± 0.9	8.9–28.4
	Inner lens surface radius	µm	27	11.0 ± 3.0	6.9–19.8
Crystalline cone	Length in LA	µm	12	6.3 ± 1.1	4.5–7.9
	Length in DA	µm	12	7.7 ± 0.8	6.6–9.8
	Distal diameter in LA	µm	16	11.4 ± 1.1	10.2–13.9
	Distal diameter in DA	µm	16	12.7 ± 1.2	10.3–14.2
Pigment granule	PPC diameter	nm	55	495.3 ± 165.0	199–922
	SPC diameter	nm	55	517.2 ± 88.3	192–867
Rhabdom	Length of peripheral rhabdomeres	µm	6	11.9 ± 0.5	11.4–12.4
	Length of central rhabdomeres	µm	17	43.5 ± 1.6	33.0–55.3
	Distal diameter	µm	32	6.7 ± 0.5	5.9–7.9
	Microvillus diameter	nm	60	36.4 ± 5.8	20–50
	Rhabdom cross-sectional area in LA	µm^2^	24	120.4 ± 18.0	90.4–184.8
	Rhabdom cross-sectional area in DA	µm^2^	18	152.1 ± 23.7	120.2–196.2
	Pigment granule diameter	nm	65	502.2 ± 83.3	179–950
Ommatrichia	Number	-	10	7.8 ± 0.5	7–12
	Length	µm	20	10.1 ± 0.5	6.0–14.7
Basal matrix	Thickness	µm	30	9.0 ± 1.7	5.8–11.5
Optical characteristics	Focal length	µm	-	31.3	-
	Image focus length	µm	-	42.2	-
	F-number	-	-	1.5	-
	Ommatidium acceptance angle	°	-	12.2	-
	Interommatidial angle	°	-	3.2	-
	Eye parameter	µm·rad	-	0.9	-

Abbreviations: OLU—outer lens unit; ILU—inner lens unit; LA—light adaptation; DA—dark adaptation; PPCs—primary pigment cells; SPCs—secondary pigment cells.

### 3.2. Internal Structures of the Eyes

The ommatidium of *T. yunnanensis* consists of two primary components: (1) the dioptric apparatus, formed by the cornea and an acone-type crystalline cone, and (2) the photosensitive layer, containing retinular cells and their rhabdomeres. Pigment cells occupy the spaces between adjacent rhabdom units, providing optical isolation for each ommatidium. Ommatidial length varies from 33.0 μm in the central region to 55.3 μm in the marginal region. The lack of a broad, clear zone identifies the eye as the apposition type. Figure 3 provides a schematic diagram of the ommatidial fine structure, based on observations using LM and TEM.

#### 3.2.1. Cornea

The cornea is the outermost structure of the ommatidium in the compound eye of *T. yunnanensis*, serving mechanical support and protective functions. Its surface is smooth and lacks corneal nipples, with a maximum thickness of 20.7 ± 1.1 µm (Figure 3B; Table 1). In longitudinal section, the corneal lens presents as a thick biconvex structure, comprised of an outer lens unit (OLU) measuring 16.0 ± 2.4 µm thick and an inner lens unit (ILU) measuring 2.4 ± 0.5 µm thick. Toluidine blue staining reveals the OLU remains unstained, while the ILU stains densely (Appendix A). TEM further demonstrates a laminar corneal structure, consisting of 67–74 layers exhibiting alternating electron densities. Transverse sections show chitin-protein microfibrils arranged in parabolic arcs; these microfibrils splay laterally and converge toward a central core corresponding to the ommatidial axis, coiling around it to form a spiral structure (Appendix A).

#### 3.2.2. Crystalline Cone

Immediately beneath each cornea lies an acone-type crystalline cone composed of four wedge-shaped cone cells, each constituting one-quarter of the entire cone (Figure 3B–D). The crystalline cone is 12.0 ± 1.3 µm at distal diameter and peripherally surrounded by pigment cells (Figure 3C,D). Additionally, the proximal end of the crystalline cone is observed to branch into several projections that penetrate the underlying rhabdom (Appendix A). Surprisingly, dense aggregations of virus-like particles (VLPs) were detected near the nuclei of cone cells in ommatidia located at the marginal region of the compound eye (Appendix A).

#### 3.2.3. Retinular Cells and Rhabdom

Each ommatidium comprises eight retinular cells forming a typical “insula-pattern” open rhabdom (Figure 3A,E–G). Within this structure, the rhabdomeres of the six peripheral cells (R1–R6), measuring 11.9 ± 0.5 µm in length, encircle the central rhabdomere formed by R7 and R8 (which is significantly longer at 43.5 ± 1.6 µm) (Table 1). Although the peripheral rhabdomeres contact each other only at their distal tips to form a complete cylinder, transverse sections reveal them as isolated units (Figure 3E–G). R1–R6 begin separating from each other approximately one-quarter of the way down from the distal tip, a level coinciding with their nuclei. In contrast, the nuclei of R7 and R8 are positioned 5–7 µm above the basal matrix; these central cells are typically closer together distally than proximally (Figure 3H). Notably, the peripheral rhabdomeres R3 and R6 are typically shorter than R1, R4, R2, and R5, with the latter four being consistent in length with the central R7/R8 rhabdomere (Figure 3I). Transverse sections further reveal regional differences: the peripheral rhabdom system exhibits square facets in the dorsal eye region and hexagonal facets in the ventral region (Appendix A), with the cross-sectional areas of the rectangular (dorsal) rhabdoms being significantly smaller than the hexagonal (ventral) types (*p* = 0.039). The microvilli of the central rhabdomere converge in one central cell and diverge in the other (Figure 3F,G). The cytoplasm of all retinular cells contains various organelles, including mitochondria, multivesicular bodies, endoplasmic reticulum, and pigment granules (Appendix A). These pigment granules have a diameter of 502.2 ± 83.3 nm, similar in size to those in the pigment cells (Table 1).

#### 3.2.4. Pigment Cells

Each ommatidium in *T. yunnanensis* possesses two primary pigment cells (PPCs) proximal to the crystalline cone and at least 17 surrounding secondary pigment cells (SPCs) (Figure 3C,D). The nuclei of the SPCs are positioned at approximately the same level, corresponding to the distal fusion point of the rhabdom (Figure 3E). Both PPCs and SPCs contain abundant spherical, electron-dense pigment granules (PGs) within their cytoplasm. The pigment granules in PPCs (diameter: 495.3 ± 165.0 nm) and SPCs (diameter: 517.2 ± 88.3 nm) exhibit similar sizes (Figure 3C; Table 1).

#### 3.2.5. Basal Matrix

The basal matrix, a multiporous structure separating the retina from the lamina, measures 9.0 ± 1.7 µm in thickness (Figure 3H–J; Table 1). It contains circular perforations through which bundles of axons, each enveloped by glial cells, pass from the ommatidia. Each axon bundle comprises eight fibers: two large-diameter fibers centrally positioned, flanked by five smaller fibers on each lateral side. These bundles traverse the basal matrix and terminate directly onto the rhabdomeres, with their termination points located 1.6 to 5.7 µm above the basal matrix (Figure 3I). Additionally, the basal matrix houses large nuclei and mitochondria (Appendix A). Tracheoles from the tracheal systems both above and below the basal matrix extend distally through spaces within the interommatidial region (Appendix A).

### 3.3. Morphological Changes in the Eye During Dark/Light Adaptation

Under light adaptation, the crystalline cone exhibited a conical morphology with a narrow, pointed proximal end, measuring 6.3 ± 1.1 µm in length and 11.4 ± 1.1 µm in diameter (Table 1). Primary pigment cells, containing larger pigment granules, surrounded the cone (Figure 4A,C,E). Mitochondria in central retinular cells were smaller than in dark adaptation. Following dark adaptation, cone dimensions significantly increased (length: *p* = 0.001; diameter: *p* = 0.003; Table 1; Figure 4A,B,F), suggesting light-induced compression or contraction. Concurrently, the rhabdom area expanded significantly from 120.4 ± 18.0 µm^2^ to 152.1 ± 23.7 µm^2^ (*p* < 0.001) (Table 1). Pigment granule counts showed no significant difference between states (*p* = 0.197), though smaller granules migrated proximally toward the cone during dark adaptation (Figure 4E,F). These modifications demonstrate adaptive plasticity in the compound eyes.

### 3.4. Optical Features of the Compound Eyes

To characterize the optical system, several key parameters were calculated (Table 1). Focal length (f) and image focal length (f’) were derived using f = n/P_1_ and f’ = n’/P_1_, where n = 1 (air) and n’ = 1.348 (image space). P_1_ = (n_1_ − n)/r_1_ (n_1_ = 1.452 lens refractive index; r_1_ = distance from corneal lens curvature center to rhabdom tip), yielding P_1_ = 0.0319, f = 31.3 µm, and f’ = 42.2 µm. This f places the focal plane near the rhabdom distal tip. The F-number (f/A, A = maximum facet diameter) was 1.49, quantifying light-gathering efficiency. Spatial resolution (interommatidial angle Δφ = A/R) was 3.2°; eye radius R was geometrically estimated (R = (s^2^/4 + h^2^)/(2h), s = segment length, h = height). The acceptance angle Δρ_rh_ (rhabdom distal diameter/f) was 12.2°. Eye parameter P = D^2^/R (D = facet diameter, R = eye radius) was 0.9 µm·rad.

### 3.5. Daily Activities Patterns of T. yunnanensis

Field observations of *T. yunnanensis* confirm exclusively diurnal flight activity. No flights occurred during night hours (Figure 5). Individuals emerge from tree trunks around 06:00, with initial flight initiation at 16 lux light intensity. Peak flight activity (07:00–11:00) coincides with increasing light intensity (305 lux to 4841 lux), while sporadic flights continue throughout daylight hours.

### 3.6. Phototactic Response of T. yunnanensis to Different Monochromatic Wavelengths

Both sexes exhibited significant broad-spectrum phototaxis across all tested wavelengths (ANOVA: females F _(27,56)_ = 6.849, *p* < 0.001; males F _(27,56)_ = 13.100, *p* < 0.001; Figure 6A,B). Post-hoc LSD tests confirmed enhanced phototactic response in full-light versus full-darkness conditions (*p* < 0.001). Sex-specific spectral sensitivity differed markedly: females showed peak sensitivity at 360 nm, 380 nm, and 700 nm wavelengths (statistically indistinguishable from natural light responses), while males demonstrated selective sensitivity only at 360 nm and 400 nm.

## 4. Discussion

While the compound eyes of Coleoptera, particularly within the Cucujiformia (including Cleroidea, Lymexyloidea, Chrysomeloidea, and Cucujoidea), have been comprehensively studied [25,27,32,33], morphological analyses of Scolytinae bark beetles remain scarce [25]. This gap is especially significant given their global diversity (>6000 species) and severe ecological impact on arboreal ecosystems. Currently, compound eye structure and phototaxis-related functions have been characterized in only fifteen Scolytinae species: five seed beetles (*Conophthorus ponderosae*, *C. teocotum*, *C. conicolens*, *C. michoacanae*, *Hypothenemus hampei*) [36,37], two ambrosia beetles (*Xyleborus ferrugineus*, *T. lineatum*) [12,13], and nine bark beetles (*Dendroctonus ponderosae*, *D. rufipennis*, *D. pseudotsugae*, *D. valens*, *I. paraconfusus*, *Dryocoetes autographus*, *Hylastes nigrinus*, *Hylurgopinus rufipes*, *Scolytus multistriatus*) [18,21,22,23,24,38,39]. However, ultrastructural analyses remain limited to two studies [12,13,36], with Mora et al.’s results exhibiting unclear imaging. Our findings provide novel insights into the ultrastructure of the compound eye and its role in phototaxis for an additional bark beetle species. This work advances the understanding of visual–olfactory integration within this ecologically critical group and informs targeted pest management strategies.

### 4.1. Morphological Features of the Compound Eyes in T. yunnanensis

Generally, insect eye size decreases with body size, a pattern evident in larger specimens across groups [16,40,41]. However, most scolytid ommatidia number shows no significant allometric scaling with body size, likely reflecting functional constraints tied to their concealed lifestyle [5,36,39]. Instead, each scolytid species typically possesses a characteristic ommatidia range, varying from ~90 (*Conophthorus*) to ~400 (*Dendroctonus*) facets per eye [36,39]. Exceptions like *T. lineatum*, *H. nigrinus*, and *H. hampei* are rare [12,13,18,37]; *T. yunnanensis* follows the common non-allometric pattern. Significant sexual dimorphism in facet number is absent in bark beetles (e.g., *D. pseudotsugae*, *D. ponderosae*, *T. yunnanensis*) but occurs in some ambrosia beetles like *H. hampei* [36,37,39]. Within species, the correlation between facet number and body length is consistently low (r ≈ 0.3–0.4) [39], as seen in *T. yunnanensis*. Minor asymmetries exist between left/right eyes (Figure 2), but scolytid eyes are universally simpler and possess far fewer ommatidia (<400) compared to other wood-associated beetles (e.g., Buprestidae and Cerambycidae) or odor-guided beetles like carrion beetle, *Necrophorus* (~3500 facets) [33,39,40]. This ocular simplicity correlates with their adult life spent concealed within wood and a primary reliance on olfactory cues for locating breeding sites. Certainly, facet count and their surface area distribution correlate with visual ability in insects, as seen in holoptic-eyed dipterans and hymenopterans where these features accelerate phototransduction [27,42], such differences require cautious interpretation due to the multifactorial nature of vision [17,28,34]. Eye orientation also influences behavior; for instance, *D. frontalis* prefers landing on vertical trees, while *Ips* species show no such preference [14,38].

Ommatidial shape exhibits distinct dorsal-ventral regionalization, transitioning between quadrilateral and hexagonal facets (Figure 1C–E). Similar regionalization occurs in various beetles, including *Creophilus erythrocephalus*, *Sartallus signatus* [43], *Neotriplax lewisi* [44], and *Monochamus alternatus* [40], though these differ externally. Unlike specialized dorsal rim ommatidia (DRM) that mediate polarization sensitivity for navigation [45], *T. yunnanensis* lacks this modification. However, in flying beetles like *T. yunnanensis* with vertically oriented eye axes, dorsal regions may still exhibit polarization sensitivity. This functional adaptation is consistent with both the eye’s perpendicular alignment to incident light and its internal rhabdomere architecture [40,44,46].

The compound eye of *T. yunnanensis* is an acone apposition eye with an open rhabdom, a type found in Diptera, Hemiptera, and some beetles like Cucujiformia [32,33,39,46]. Each ommatidium features two central retinula cells (R7, R8) largely separated from six peripheral cells (R1–R6). The peripheral rhabdomeres typically form a sleeve-like cylinder around the central pair, corresponding to the “insula-pattern with subpattern t” described by Schmitt et al. (1982) [33], a configuration associated with a day-active lifestyle [46,47]. Structurally, *T. yunnanensis* most closely resembles Galerucinae beetles and converges with the pattern in *Stenocorus* and *Stenopterus* [32,33]. Notably, this specific insula-pattern rhabdom appears to be nearly ubiquitous among scolytid species examined to date [12,13,36], highlighting a striking uniformity within Scolytinae. This stands in contrast to the considerable diversity of rhabdom types among cerambycid beetles [32,33,40,48], which include multiple patterns like insula and ponticulus, each containing several subtypes.

The cornea of *T. yunnanensis* is notably thick and laminated, exhibiting a helical arrangement of chitin and protein fibrils [43] that may refract obliquely incident light [12,13]. This robust structure likely provides enhanced protection within its boring habitat and attenuates light intensity reaching the photoreceptors, reducing photic damage risk [49]. The smooth, head-flush eye surface minimizes sunlight exposure. Interfacetal corneal pigment, combined with strong frontal facet curvature, probably filters off-axis and stray light. Furthermore, the pronounced curvature of the inner corneal lens units, directly connected to the crystalline cone, effectively guides light. The acone-type crystalline cone has optically clear cytoplasm, enabling strong refraction by the convex inner corneal surface. A small corneal cone funnels light with the corneal lens towards the distal tips of the rhabdomeres. Primary pigment cells enveloping the four cone cells regulate light reaching the rhabdom via aperture adjustment. Each ommatidium is surrounded by at least 17 secondary pigment cells (Figure 3E) performing multiple functions: light screening, providing nutrition and metabolic support, structural support, and facilitating transmission of receptor potentials within the extracellular space [44].

Microvilli of R7 and R8 converge centrally, while those of R1-R6 diverge. This arrangement potentially adapts the eye for polarized light detection [32,33]. High polarization sensitivity requires photopigment chromophores within microvilli to be aligned perpendicularly to incident light direction—an alignment observed in *T. yunnanensis*. Consequently, retinula cells with a single microvillar alignment direction are presumed to be polarization sensitive. In such systems, electrical coupling between cells would likely decrease spectral and polarization sensitivity while increasing absolute sensitivity [45]. This could enable these cells to function as highly efficient lateral filters, maximizing their functional advantage [33,47].

### 4.2. Light-Adaptive Modifications in the Ommatidia of T. yunnanensis

Variations in light conditions across insect lifestyles and habitats necessitate adaptive photoreceptor structural changes [46]. A key adaptation in many insects involves pigment granule migration within primary pigment cells (PPCs), widening or narrowing the ommatidial aperture to regulate light entry—a typical “pupil” mechanism [44]. Like other insects (e.g., *Creophilus erythrocephalus* [43], *Xanthochroa luteipennis* [44], *M. alternatus* [40], *Polyrhachis Sokolova* [50]), *T. yunnanensis* shows minimal pigment granule displacement under light adaptation. In the dark-adapted states, granules migrate proximally near the cone tip, minimizing light obstruction. The cone lacks crystalline material, indicating weak intrinsic light-gathering capacity [32]. However, its shape changes dynamically: shortening and widening under bright light versus elongating in dim light. This deformation, driven by microtubular organelles (not muscle fibers) [44], alters the light path and reduces absorption by screening pigments. The cross-sectional area and shape of the rhabdoms adapt to control light flux and maximize photopigment interaction. Such rhabdom volume changes are widespread in apposition eyes coping with light variation [50,51]. During daytime, vision primarily involves central rhabdomeres. At night, peripheral rhabdomeres may also contribute due to increased pupil aperture. If central and peripheral retinula cells possess different spectral sensitivity maxima (as in other open-rhabdom insects [47,51]), this nocturnal shift could broaden spectral acceptance, enhancing photon capture efficiency compared to monochromatic tuning [29,44,51]. This mechanism is commonly used in the transition between dark and light adaptation in order to adapt apposition eyes to different light sensitivities [25,51,52]. Therefore, *T. yunnanensis* predominantly relies on crystalline cone and rhabdom plasticity rather than pronounced pigment migration, representing a distinct adaptation strategy where pigment movement serves a secondary role.

### 4.3. Optical Properties, Phototactic Behavior, and Spectral Sensitivity in T. yunnanensis

According to Stavenga (2003) [35], insect eyes with F-numbers below 2 possess rhabdoms that function as waveguides, where a smaller F-number corresponds to higher absolute optical sensitivity, while a larger F-number indicates reduced photosensitivity. Nocturnally active insects with apposition eyes typically exhibit F-numbers ranging from 0.5 to 1.2, whereas diurnal species display higher values [53]. In *T. yunnanensis*, the calculated F-number of 1.5 suggests relatively low absolute sensitivity to light reaching the target rhabdom via its facet. This aligns with our observations of the species’ diurnal activity patterns (Figure 5).

Furthermore, a low eye parameter (P) signifies adaptation to well-illuminated environments. Diurnal insects with apposition compound eyes typically exhibit P-values between 0.25 and 2 μm·rad [40,52,54]; our calculated value of 0.9 μm·rad for *T. yunnanensis* falls within this diurnal range and is consistent with experimental observations. For example, our behavioral records showed that most individuals remain in the stationary zone, while a small portion disperses toward either the light zone or dark zone, with nearly equal numbers in both areas. This phenomenon is also observed in other beetles such as *Pissodes punctatus* [55]. This confirms that the eyes of *T. yunnanensis* are well adapted to diurnal photic conditions. Regarding spatial resolution, the interommatidial angle (Δφ) of *T. yunnanensis* is 3.2°, a relatively wide value typical of herbivorous, non-fast-flying insects. While larger Δφ generally correlates with lower resolution, this angle likely provides sufficient resolving power for the species’ ecological needs. The ratio of the acceptance angle (Δρ) to the interommatidial angle (Δρ/Δφ = 3.8) indicates significant oversampling of the visual field, prioritizing image brightness over maximum spatial resolution [34]. Combined with the low eye parameter P, this adaptation may be particularly advantageous for detecting specific light wavelengths, such as color changes in canopy profiles.

In Cucujiformia beetles, compound eyes have lost the ancestral blue-sensitive opsin but recurrently evolved trichromacy through duplications of UV-sensitive and long-wavelength (LW) opsin genes [25]. These opsins form optical waveguides sensitive to near-UV (300–400 nm) and long-wavelength light (400–620 nm) [22]. While species like *D. pseudotsugae* and *I. paraconfusus* exhibit peak sensitivities to blue (λ_max_ = 450 nm) and green light (λ_max_ = 520 nm) [22], our phototaxis and spectral sensitivity data reveal that *T. yunnanensis* possesses another two photoreceptor classes [25]: responding to UV and red light. This aligns with certain Coleoptera species (e.g., *Tribolium castaneum* [56], *Luciola cruciata* [57]) that lack blue opsins. *T. yunnanensis* exhibits a strong behavioral preference for UV and red-light stimuli (Figure 6). Its UV sensitivity aligns with patterns observed across diverse beetles [25,56,57], enabling sky polarization detection and solar navigation for flight stabilization [44]. This adaptation is ecologically critical given its high-altitude habitat on the Yunnan-Guizhou Plateau, where intense UV radiation likely guides host-tree selection during peak flight activity. Furthermore, female-specific sensitivity to 700 nm red light may serve as a visual cue for pioneer individuals to identify weakened *P. yunnanensis* hosts, consistent with the reddish hues of deteriorating pine canopies (see Figure 1 in Cui et al. [6]). This functional interpretation parallels findings in *Acyrthosiphon pisum*, where positive phototaxis in early-stage adults facilitates vertical movement to plant apices for population dispersal [58]. These phototactic behaviors suggest potential applications for controlling *T. yunnanensis* populations. Light-based trapping of flight forms could limit damage [17,29], though future studies should first confirm whether dedicated UV- and red-sensitive photoreceptors mediate these responses.

## 5. Conclusions

Light detection and corresponding photobehaviors represent fundamental characteristics in insects, including concealed species like *T. yunnanensis*, critically guiding pest control strategies and providing foundational principles for genetic research. This study establishes that *T. yunnanensis* possesses an acone apposition compound eye with an open-rhabdom configuration. Quadrilateral facets occupy the dorsal third of the eye, while hexagonal facets dominate the remaining regions. Dynamic structural changes in ommatidia during dark/light adaptation demonstrate optimization for varying ambient light conditions—an adaptation to its high-light plateau environment. Behavioral observations confirm peak flight activity occurs exclusively between 07:00–11:00 h with no nocturnal activity. Phototaxis assays revealed sexually dimorphic responses: females exhibit high sensitivity at 360 nm, 380 nm, and 700 nm; males show peak sensitivity at 360 nm and 400 nm. This work advances beetle visual ecology by demonstrating how *T. yunnanensis* integrates spectral sensitivity with olfactory cues for host location and non-host avoidance. These findings enable globally applicable strategies to protect pine forests against mass *Tomicus* outbreaks.

## Figures and Tables

**Figure 3 biology-14-01032-f003:**
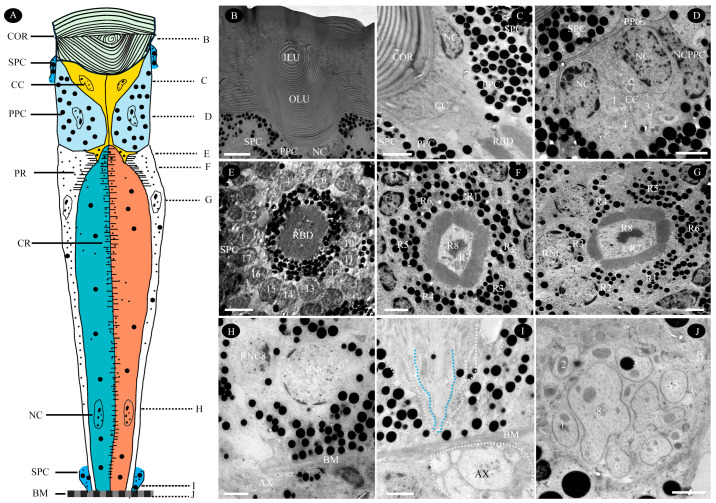
Open rhabdoms and their corresponding structures in the ommatidium of *T. yunnanensis*. (**A**) Schematic diagram of an ommatidium. Left abbreviations denote structures; right letters (**B**–**J**) indicate corresponding sections. (**B**) Corneal longitudinal section showing outer/inner lens units with differing electron densities. (**C**) Crystalline cone longitudinal section revealing four cone cell shapes and their association with the distal rhabdom. (**D**) Crystalline cone transverse section displaying four numerically labeled cone cells. (**E**) Distal rhabdom transverse section: completely fused, circular rhabdom surrounded by at least 17 numbered secondary pigment cells. (**F**) Rhabdom transverse section showing insula pattern: central rhabdomeres 7/8 and peripheral rhabdomeres 1–6. (**G**) Rhabdom transverse section with three visible retinular cell nuclei. (**H**) Transverse section through central rhabdomeres 7/8: two nuclei near basal membrane; axon bundle penetrating membrane. (**I**) Proximal ommatidium longitudinal section: six retinular cells (central cyan-dotted, peripheral white-dotted) reach basal matrix and connect to axons. (**J**) Axon bundle transverse section showing eight axons per bundle within glial sheaths, arrangement non-corresponding to retinular sequence. AX—axon; BM—basal matrix; CC—cone cell (yellow); COR—cornea (green); CR—central rhabdomere; ILU—inner lens unit; NC—nucleus; NCPPC—nucleus of primary pigment cell; OLU—outer lens unit; PPC—primary pigment cell (light blue); PR—peripheral rhabdomere; R1–8—retinular cell numbers 1–8, the labeled orange and cyan regions represent the position of retinular cell 7 and 8, respectively; RBD—rhabdom; RNC—retinular cell nucleus; SPC—secondary pigment cell. Scale bar: (**B**,**E**) = 5 µm; (**C**,**D**,**F**–**I**) = 2 µm; (**J**) = 1 µm.

**Figure 4 biology-14-01032-f004:**
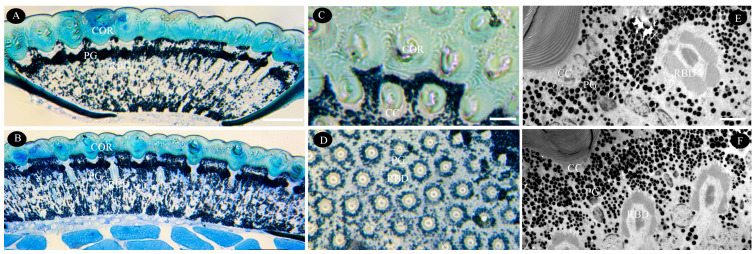
Morphological changes in *T. yunnanensis* ommatidia during dark/light adaptation. (**A**,**B**) Light micrographs showing overall compound eye structure under light (**A**) and dark (**B**) adaptation. (**C**,**D**) LM details of the dioptric apparatus (**C**) and the photosensitive layer (**D**) under light adaptation. (**E**,**F**) TEM micrographs of pigment granule distribution between the dioptric apparatus and photosensitive layer under light (**E**) and dark (**F**) adaptation. LM images (A-D) show tissues stained with 1% toluidine blue and imaged under white light. CC—cone cell; COR—cornea; DA—dioptric apparatus; PG—pigment granules; PSL—photosensitive layer; RBD—rhabdom. Scale bar: (**A**,**B**) = 50 µm; (**C**,**D**) = 20 µm; (**E**,**F**) = 5 µm.

**Figure 5 biology-14-01032-f005:**
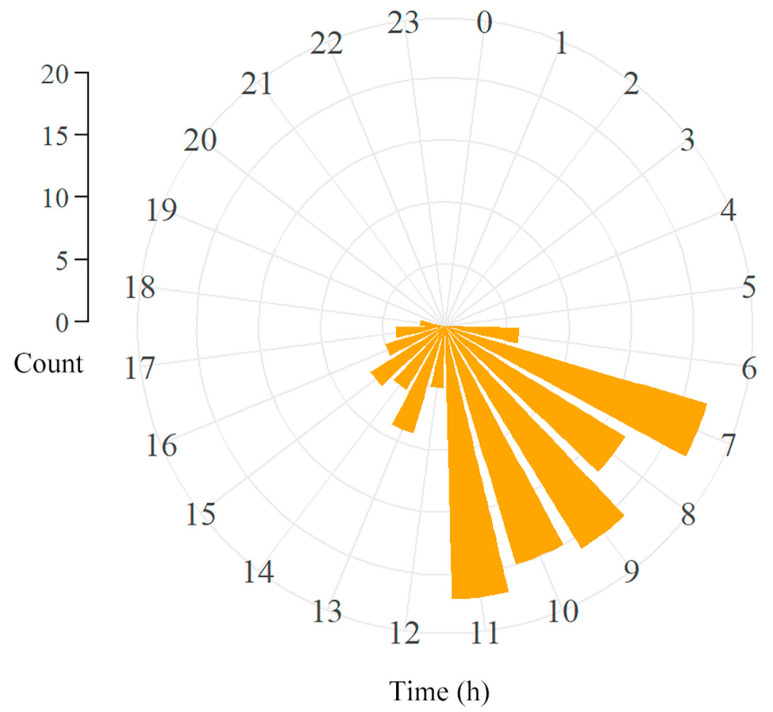
Hourly flight initiation events of *T. yunnanensis* in natural photoperiod (light phase: 07:30–19:30; dark phase: 19:30–07:30 local time).

**Figure 6 biology-14-01032-f006:**
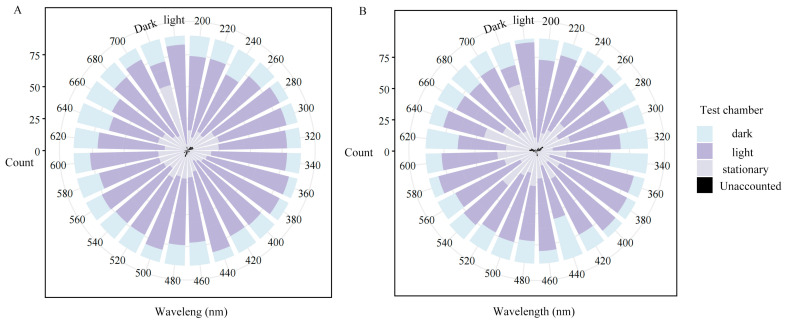
Phototactic responses of *T. yunnanensis* to darkness, natural light, and monochromatic wavelengths (200–700 nm). (**A**) Females; (**B**) males.

## Data Availability

The original contributions presented in this study are included in the article/Appendix A. Further inquiries can be directed to the corresponding authors.

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
