# Peer review of "Compound Eye Structure and Phototactic Dimorphism in the Yunnan Pine Shoot Beetle, *Tomicus yunnanensis* (Coleoptera: Scolytinae)"

_biology, 2025, doi:10.3390/biology14081032_

Round 1
Reviewer 1 Report
Comments and Suggestions for Authors
Your manuscript is well-organized and presents novel, valuable findings on sexually dimorphic phototaxis and compound eye structure in Tomicus yunnanensis. The integration of detailed microscopy with behavioral analyses significantly strengthens the study’s contribution to insect sensory biology. Most suggested revisions involve straightforward clarifications or expansions in the discussion. Addressing these points will improve the clarity, rigor, and overall impact of your work.

Author Response
|
1. Summary |
|
|
|
Thank you for your valuable feedback and suggestions regarding our manuscript (biology-3787319). We have fully incorporated all of your recommendations. All modifications made in response to your comments are clearly highlighted in red within the resubmitted manuscript files. We have also specifically addressed the two key points raised by the reviewers as follows:
(1) straightforward clarifications: We have conducted a comprehensive revision of the manuscript, carefully addressing the reviewer comments provided within each section, particularly focusing on the Introduction and Materials and Methods. These updated descriptions and supplemental materials significantly enhance the clarity, rigor, and overall quality of the manuscript.
(2) expansions in the discussion. We have expanded the discussion regarding the number of facets and differences in their surface area distribution, which was omitted from the previous manuscript. This addition leads to a more accurate and focused discussion of our results and facilitates comparisons with similar studies.
|
||
|
2. Questions for General Evaluation |
Reviewer’s Evaluation |
Response and Revisions |
|
Does the introduction provide sufficient background and include all relevant references? |
Can be improved |
We have expanded the Introduction to include essential background research highlighting the lack of data on the species’ visual anatomy and behavior. These additions are highlighted in red within the resubmitted manuscript. |
|
Are all the cited references relevant to the research? |
Yes |
While reviewers did not request changes to references, we conducted a thorough review of all cited sources central to our research. |
|
Is the research design appropriate? |
Yes |
No revisions were required for this point. |
|
Are the methods adequately described? |
Can be improved |
Surely, we have clarified potentially confusing descriptions within the Materials and Methods. Please refer to the highlighted sections in the updated manuscript. |
|
Are the results clearly presented? |
Yes |
While reviewers made no specific requests for the Results section, we have clarified potentially confusing descriptions to enhance precision. Please refer to the highlighted text in the updated manuscript. |
|
Are the conclusions supported by the results? |
Can be improved |
Following the reviewer’s advice, we have focused on the key applications on pest control and future genetic studies. Please check the sections highlighted in red within the updated manuscript. |
|
Are all figures and tables clear and well-presented? |
Can be improved |
We have conducted a thorough review of all figures and tables, correcting identified errors in the resubmitted manuscript. Specifically, the thickness values for ILU and OLU in Figure 3B and Table 1 have been revised to rectify labeling inaccuracies resulting from measurement omissions. We confirm that optical parameters remain unchanged, as they are calculated from corneal curvature radii (external and internal measurements), which remain unaffected by these thickness adjustments. Please verify these corrections in the updated manuscript. |
|
3. Point-by-point response to Comments and Suggestions for Authors |
||
|
Comments 1: Line 2, Consider clarify the focus on sex differences in the title as another way because in this case, sexual dimorphism is a key finding. So readers and reviewers can immediately recognize its significance.
|
||
|
Response 1: Thank you for pointing this out. We fully agree and have therefore revised the title to better highlight the phonotactic differences between sexes—a central finding in our study on scolytid beetles. Please review the updated version in the resubmitted manuscript.
|
||
|
Comments 2: Line 20, That sounds slightly exaggerated. I recommend more accurate wording like 'exhibited significant phototactic responses to |
||
|
Response 2: Agree. We have revised the previously overstated descriptions to use objective terminology throughout the manuscript. These updates are highlighted for your review in the updated version.
Comments 3: Line 70, The introduction could better emphasize the research gap regarding the role of vision. So the authors may add a sentence explicitly stating the lack of data on the species’ visual anatomy and behavior. Response 3: Okay, we have added the following statement: "However, the potential role of vision in host location remains largely unexplored. Crucially, fundamental data on the species' visual anatomy (including compound eye structure) and light-guided behavioral responses are lacking, despite evidence supporting visual capabilities in other Scolytinae beetles." These additions explicitly underscore the knowledge gap in this species' visual ecology. Please review the updated manuscript. Comments 4: Line 100, Please describe host/tree selection and extraction method. Response 4: Thank you for this valuable feedback. We have enhanced the manuscript by incorporating detailed descriptions of host tree developmental stages and citing Cui et al.'s research (Insect 2023, 14(12):933), which documents representative symptoms. Additionally, we provide supplementary images (Figure 1) confirming these same features, consistent with Cui et al.'s findings. Figure 1 Trees in early infestation stages or predisposed to attack exhibit boreholes and resin exudation on shoots/trunks, crown dehydration, and progressive wilting.
Comments 5: Line 106, Please provide more detail on the sexing procedure, including tools used and whether stridulation was systematically or occasionally used. Response 5: As only male T. yunnanensis produce chirping sounds, acoustic signals served as our primary sex identification method. To account for occasional non-vocalizing males, we implemented morphological validation by examining the last abdominal tergites under a Zeiss Discovery V20 stereomicroscope. Females exhibit arched tergites while males show rectangular morphology (Figure 2). These methodological clarifications, along with a newly added supporting reference, appear in the revised manuscript. Figure 2 The external shape of abdominal terminal tergum in adult T. yunnanensi (A, female; B, male)
Comments 6: Line 111, In this case, these fixation conditions may introduce artifacts. Please justify their use. Response 6: Thank you for highlighting this concern. To prevent experimental artifacts, all dark-environment trials were conducted in a custom-built soundproof chamber (2m × 2m × 3m). This optically sealed laboratory space achieves complete darkness when sealed, enabling uncontaminated chirp recording without external noise interference. Specifically designed for beetle bioacoustics research, the chamber also provides ideal dark-adapted conditions for studying compound eye physiology. We have detailed this experimental setup in the revised manuscript. Please check the revised manuscript.
Comments 7: Line 115-116, Consider whether sample size was sufficient for detecting sex-based differences. Response 7: Except for SEM observations, earlier light microscopy examinations revealed no sexual dimorphism in ommatidia number, aligning with Chapman's findings in Dendroctonus species (Ommatidia numbers and eyes in scolytid beetles. Ann Entomol Soc Am, 65(3), 550-553). As both Dendroctonus and Tomicus belong to the Hylurgini tribe (Scolytinae), this provides indirect evidence for ommatidia patterns. Moreover, while existing studies on beetle compound eye morphology typically sample ≤20 individuals (e.g., Chu et al., 1974; Chapman, 1972; Vega et al., 2014), our analysis of 25 specimens (10♂, 15♀) exceeds this norm. This sample size not only represents the most comprehensive dataset in current literature but also explains facet variations, as detailed in our results. We believe this addresses your concerns. Do you agree with our reply?
Comments 8: Line 117, Critical parameters (e.g., duration, intensity/frequency) are not provided. So please include specifics about ultrasonic cleaning settings. Response 8: Ultrasonic cleaning provides a straightforward method for removing ocular surface contaminants from T. yunnanensis. Here, a commercially available cleaner operating at default settings effectively eliminated these impurities, as confirmed by the cleared specimen surfaces in Figure 1B. This methodological clarification appears in the revised manuscript. Thanks.
Comments 9: Line 131, which one? please quote that work! Response 9: We have updated the text to "as described in the SEM protocol" in the revised manuscript. Please review this change. |
||
|
Comments 10: Line 150, Not sure, how many beetles per replicate were used? Response 10: Our aims record beetle exit counts from stems at 60-minute intervals during both light and dark phases. We realize that the original description caused some confusion and have clarified this in the resubmitted manuscript. Please review these revisions.
Comments 10: Line 178,......Wachmann [31],.... Please recheck the reference style Response 10: Okay, this has been corrected in the revised manuscript.
Comments 11: Line 178, Please recheck the reference in style. Response 11: Okay, this has been corrected in the revised manuscript. Thanks.
Comments 12: Line 209, Expand in the discussion on what this asymmetry could mean functionally. Response 12: Okay, we have expanded the Discussion section to address the functional implications of facet number and surface area variations. These morphological differences likely represent adaptations to ecological demands—potentially enhancing light capture within complex coniferous needle arrangements or optimizing behaviors against bark substrates. However, the underlying mechanisms driving this asymmetry remain unresolved. Future genetic investigations will be essential to elucidate these adaptive traits. We trust this clarification addresses your concerns. Thanks.
Comments 13: Line 353, Add “mean ± SD” and “n=3” if applicable. Response 13: Here, hourly counts of emerging T. yunnanensis individuals at nylon bag entry points were recorded across three replicate bags at 60-minute intervals under natural photoperiod. Data represent total emergence events per hourly period. We have clarified these methodological points in the revised Materials and Methods section. Please review these updates in the manuscript.
Line 395, Line 395, carrion beetle, Necrophorus...... We acknowledge this point and have corrected the description. Kindly evaluate the revisions in the resubmitted manuscript.
Comments 15: Line 402, ....in various beetle species, including.... Response 15: We agree and have revised the description accordingly. Please review the updated manuscript.
Comments 16: Line 520, Emphasize key applications (e.g., pest control tools or future genetic studies). Response 16: We agree the conclusion should align with our paper's purpose and have revised it to: “critically guiding pest control strategies and providing foundational principles for genetic research.” Please evaluate this change in the resubmitted manuscript. Tanks.
|
||

Reviewer 2 Report
Comments and Suggestions for Authors
Dear authors,
Your manuscript is a well-structured, complete and methodologically sound study. You have successfully achieved your objective of identifying the peculiarities of the eye structure of the bark beetle T. yunnanensis, as well as describing the ultrastructure of the eyes of adult beetles. The features you have identified are undoubtedly important in general biological terms, as well as for further studies into the mechanism by which the bark beetle chooses its host plant and the mating processes of adult beetles. Overall, the work is methodologically sound, drawing comparisons with existing data on other species and providing detailed descriptions of the visual features of T. yunnanensis. However, we would like to draw your attention to the following points:
- Section ‘2.5 Observation of daily behavioural rhythms’ states that ‘Daily activity of T. yunnanensis was observed under normal laboratory room conditions’. The paper does not say how the results of the laboratory study were correlated with the daily activity of beetles in nature, as there were no controls in the study.
- In section ‘2.6 Testing phototaxis with monochromatic light across multiple wavelengths’ it is stated that: ‘All experiments occurred in a self-built darkroom’. That is, these experiments were also performed only in a laboratory environment. Was there an opportunity to determine in nature how capable beetles are of finding host plants in mixed forests? How does the spectrum of different conifers differ, and how does it relate to the wavelengths that beetles are able to capture?
- Of course, a detailed comparison of the eye structure of T. yunnanensis with close species of bark beetles was expected, but there is only a general comparison with different beetle species of individual structures in the discussion. Such a comparison would probably need to be made in the future. It would be important for revealing the mechanisms of tree colonisation by bark beetles.
Despite the above questions, the work you have submitted can be recommended for publication. It is a logically structured scientific work performed at a high methodological level. The results of the research can be used in pest control studies. Also, the work has a good prospect for further research development.
Author Response
|
1. Summary |
|
|
|
Thank you for reviewing our manuscript (biology-3787319). We have carefully addressed all reviewer comments and incorporated every recommendation. Our point-by-point responses to your concerns are detailed below, with corresponding revisions highlighted in red in the resubmitted manuscript files.
|
||
|
3. Point-by-point response to Comments and Suggestions for Authors
|
||
|
Comments 1: Section ‘2.5 Observation of daily behavioural rhythms’ states that ‘Daily activity of T. yunnanensis was observed under normal laboratory room conditions’. The paper does not say how the results of the laboratory study were correlated with the daily activity of beetles in nature, as there were no controls in the study. |
||
|
Response 1: Thank you for your feedback. The original description of “normal laboratory room conditions” meant that specimens containing stems were kept in our laboratory room, where temperature and humidity were controlled (e.g., using air conditioning). We recognize this caused confusion. We have therefore revised the text to: “in a laboratory under ambient room temperature and humidity conditions in Kunming during March”. Further clarification, Kunming (laboratory site) and Qujing (specimen collection site) share the same time zone (UTC+7) and are separated by only ~145 km. This minimal distance ensures consistent daily light cycles, meaning observed daily rhythms reflect natural emergence timing. We have revised the text accordingly. Please review this change at Lines 156-158 in the updated version.
|
||
|
Comments 2: In section ‘2.6 Testing phototaxis with monochromatic light across multiple wavelengths’ it is stated that: ‘All experiments occurred in a self-built darkroom’. That is, these experiments were also performed only in a laboratory environment. Was there an opportunity to determine in nature how capable beetles are of finding host plants in mixed forests? How does the spectrum of different conifers differ, and how does it relate to the wavelengths that beetles are able to capture? |
||
|
Response 3: Our current objective was to test phototactic responses of T. yunnanensis to monochromatic light. This aims to identify the optimal wavelength for attraction, thereby informing integrated pest management strategies. We selected wavelengths covering the known visual sensitivity range of scolytid beetles (Groberman et al., 1982; Strom, 2001; Campbell et al., 2006; van der Kooi et al., 2021). These wavelengths correspond to perceptible color shifts in pine needles, as illustrated below. Crucially, our findings suggest color serves as a key guiding cue for host selection. Although this laboratory-based study requires field validation, the alignment between observed attraction spectra and natural pine needle phenology provides a foundation for developing optical pest management tools. As a preliminary investigation, this work identifies critical visual cues and establishes a basis for future research into photoreceptors and genetic mechanisms. To improve clarity and accuracy, we have updated these descriptions in the manuscript. Your review of these changes would be greatly appreciated. Thank you.
Figure 1 Needle color and length variation in pine shoots during early infestation (Upper right) and predisposed stages (Left bottom) of T. yunnanensis. Central needles exhibit reduced length and distinct coloration (right) compared to peripheral needles.
Comments 3: Of course, a detailed comparison of the eye structure of T. yunnanensis with close species of bark beetles was expected, but there is only a general comparison with different beetle species of individual structures in the discussion. Such a comparison would probably need to be made in the future. It would be important for revealing the mechanisms of tree colonisation by bark beetles. Response 3: Comparative studies across coleopteran species, particularly scolytids, are essential to elucidate bark beetle host colonization mechanisms. However, research remains limited: only two ultrastructural studies exist on scolytids (Chu et al. 1975, 1976; Mora et al. 2014), and Mora et al.'s images lack clarity. To address this gap, we expanded our taxonomic scope to include Cucujiformia beetles (Line 381-392), establishing comprehensive morphological and functional baselines for T. yunnanensis, including ocular morphology, internal anatomy, optical properties, phototactic behavior, and spectral sensitivity. As the reviewer noted, broader phylogenetic comparisons will be needed in future work. Our ongoing work will characterize ultrastructure in sympatric Yunnan scolytids (e.g., T. minor, T. brevipilosus, T. armandi) and opsin receptors to elucidate Tomicus host selection mechanisms. Following the reviewer's suggestions, we have revised the Discussion section. Please review the marked changes in the updated manuscript. We appreciate your feedback.
|
||

Reviewer 3 Report
Comments and Suggestions for Authors
1. The title of the manuscript fully corresponds to the subject of the Biology.
2. The abstract meets the requirements and contains information about the conducted research, relevance and main results of the work.
3. The introduction meets the requirements of the journal, gives the reader an idea of Tomicus yunnanensis, its systematic position, general features of the biology of the functional role in forest ecosystems.
I have a comment on the wording of the goal and objectives of the study (L 88-95). For works on the morphology of insects, the goal and objectives formulated by the authors are too detailed, the objectives contain a list of the methods used. I think the authors need to reformulate the goal and objectives in a general form, excluding the list of methods in the research objectives.
4. The "Material and Methods" section is well described, gives an idea of the research methods used, as well as statistical methods for data processing. This section does not cause any comments.
5. The "Results" section consistently describes the data obtained, contains illustrative and tabular data. There are no comments on the design.
6. The "Discussion" section is well written, contains a comparative analysis with the available literature data, which shows a good knowledge of the material on the research topic.
7. In the "Conclusion" section, it is necessary to add several sentences on the prospects for using the obtained data, in terms of protecting forests from mass pests, insect outbreaks, etc.
In general, the manuscript fully complies with the requirements of the Biology. It contains new data for science on the morphology of insects and makes a significant contribution to the development of general entomology. It is shown that the use of modern optical research methods allows one to describe the structure and morphology of the eye of insects. Important data on the light taxis and chemocommunication of Tomicus yunnanensis were obtained, which can be used to develop methods for protecting forest ecosystems from the mass reproduction of bark beetles.
Author Response
|
1. Summary |
|
|
|
We thank you for your insightful recommendations regarding our manuscript (biology-3787319). We have incorporated all the suggested revisions comprehensively. All changes made in response to reviewer comments appear in red text in the resubmitted manuscript. Below, we provide point-by-point responses to each concern raised.
|
||
|
2. Questions for General Evaluation |
Reviewer’s Evaluation |
Response and Revisions |
|
Does the introduction provide sufficient background and include all relevant references? |
Can be improved |
Following the reviewer's suggestion, we have revised the Introduction to address gaps in visual anatomy and behavior research among scolytid beetles. Please review these changes at Lines 73-77. |
|
Are all the cited references relevant to the research? |
Yes |
No revisions were required for this point. |
|
Is the research design appropriate? |
Yes |
No revisions were required for this point. |
|
Are the methods adequately described? |
Yes |
No revisions were required for this point. |
|
Are the results clearly presented? |
Yes |
No revisions were required for this point. |
|
Are the conclusions supported by the results? |
Can be improved |
To enhance precision, we refined the Conclusion to focus on pest management implications and future genetic studies. See revisions at lines 539-540. |
|
Are all figures and tables clear and well-presented? |
Yes |
No revisions were required for this point. |
|
3. Point-by-point response to Comments and Suggestions for Authors |
||
|
Comments 1: The title of the manuscript fully corresponds to the subject of the Biology. |
||
|
Response 1: Thank you for your positive assessment. To clarify intersexual differences in phototactic behavior, the title now changes "Morphology of compound eyes and phototactic dimorphism in the Yunnan pine shoot beetle (Tomicus yunnanensis, Coleoptera: Scolytinae)". We welcome your review of this update.
|
||
|
Comments 2: The abstract meets the requirements and contains information about the conducted research, relevance and main results of the work. |
||
|
Response 2: Thank you for your positive assessment.
Comments 3: The introduction meets the requirements of the journal, gives the reader an idea of Tomicus yunnanensis, its systematic position, general features of the biology of the functional role in forest ecosystems. Response 3:
Comments 4: I have a comment on the wording of the goal and objectives of the study (L 88-95). For works on the morphology of insects, the goal and objectives formulated by the authors are too detailed, the objectives contain a list of the methods used. I think the authors need to reformulate the goal and objectives in a general form, excluding the list of methods in the research objectives. Response 4: Okay, we acknowledge the original goal statement contained unnecessary details. We have therefore revised the research objectives to eliminate methodological listings. The updated text appears at lines 91-94. Thanks a lot.
Comments 5: The "Material and Methods" section is well described, gives an idea of the research methods used, as well as statistical methods for data processing. This section does not cause any comments. Response 5: Thank you for your positive assessment.
Comments 6: The "Results" section consistently describes the data obtained, contains illustrative and tabular data. There are no comments on the design. Response 6: Thank you for your positive assessment.
Comments 6: The "Discussion" section is well written, contains a comparative analysis with the available literature data, which shows a good knowledge of the material on the research topic. Response 7: Thank you for your positive assessment.
Response 8: In the "Conclusion" section, it is necessary to add several sentences on the prospects for using the obtained data, in terms of protecting forests from mass pests, insect outbreaks, etc. Comments 8: Tomicus beetles pose a significant global threat to coniferous forests, occurring widely across the Northern Hemisphere (particularly in Europe and Asia) with expanding invasions in North Africa and North America. This necessitates developing targeted management strategies to protect forest ecosystems. Following the reviewer's suggestion, we have incorporated this contextual statement at lines 549-550. Thanks a lot.
|
||

Round 2
Reviewer 1 Report
Comments and Suggestions for Authors
The revised manuscript demonstrates improved clarity and structure, making it more accessible and informative for readers. The study provides valuable insights, particularly for entomologists, and serves as a useful reference point for future research on other insect taxa. I encourage further support for this type of detailed anatomical and behavioral investigation, which contributes meaningfully to the field of insect sensory biology and forest pest management.

Author Response
We are deeply grateful for your thorough and insightful review of our manuscript (biology-3787319). The reviewer suggestions have been fully implemented, with revisions clearly highlighted in red in the resubmitted files. Below, we provide point-by-point responses to each concern raised.
Comments 1: Line 40, Consider changing the phrase to...for host location and non-host avoidance.
Response 1: We fully agree with this suggestion, which enhances scientific precision. The revision has been implemented at lines 40-41 and is highlighted in red in the updated manuscript. Thanks a lot.
Comments 2: Line 79, Dendroctonus pseudotsugae.
Response 2: This change has been incorporated into the revised manuscript. Kindly review the updated content.
Comments 3: * denotes abbreviations used in this column.. So please recheck!!
Response3: We agree the original description was unnecessarily complex and have simplified it in the revised manuscript. Thank you for your valuable input.
Comments 4: dark/light is more common in visual physiology literature.
Response 4: We accept this recommendation and have revised the manuscript accordingly. Please verify the modification at your convenience.
Comments 5: Dendroctonus ponderosae
Response 5: We agree and have incorporated this change in the revised manuscript. Thanks a lot.
Comments 6: .....the understanding.....
Response 6: This was an oversight on our part. We have addressed the omission and incorporated these revisions in the updated manuscript. Thank you for highlighting this.
Comments 7: Repetitive use of "Furthermore, Furthermore"
Response 7: Agreed. We have removed this redundancy in the revised manuscript. Kindly confirm the change upon review. Thank you for your guidance.
